# A causal role for the right frontal eye fields in value comparison

Ian Krajbich[1]*[†], Andres Mitsumasu[2][†], Rafael Polania[2,3], Christian C Ruff[2‡], Ernst Fehr[2‡]

[1]Departments of Psychology, Economics, The Ohio State University, Columbus, United States; [2]Zurich Center for Neuroeconomics, Department of Economics, University of Zurich, Zurich, Switzerland; [3]Decision Neuroscience Lab, Depterment of Heatlh Sciences and Technology, ETH Zurich, Zurich, Switzerland

**Abstract** Recent studies have suggested close functional links between overt visual attention and decision making. This suggests that the corresponding mechanisms may interface in brain regions known to be crucial for guiding visual attention – such as the frontal eye field (FEF). Here, we combined brain stimulation, eye tracking, and computational approaches to explore this possibility. We show that inhibitory transcranial magnetic stimulation (TMS) over the right FEF has a causal impact on decision making, reducing the effect of gaze dwell time on choice while also increasing reaction times. We computationally characterize this putative mechanism by using the attentional drift diffusion model (aDDM), which reveals that FEF inhibition reduces the relative discounting of the non-fixated option in the comparison process. Our findings establish an important causal role of the right FEF in choice, elucidate the underlying mechanism, and provide support for one of the key causal hypotheses associated with the aDDM.

*For correspondence:
krajbich@gmail.com

[†]These authors contributed equally to this work
[‡]These authors also contributed equally to this work

Competing interest: The authors declare that no competing interests exist.

## Editor's evaluation

Previous studies have suggested that the fixation of an object enhances the gain of its value signals that are temporally integrated during deliberation. The authors of the present study demonstrated that inhibition of the right frontal eye field (FEF) with transcranial magnetic stimulation reduces this multiplicative effect of fixation, suggesting that the FEF might be involved in the gaze-dependent modulation of value signals during decision making.

## Introduction

Despite the fact that decision making and visual attention are both central features of cognition, we still know relatively little about how they interact. A prominent view in decision neuroscience is that the decision process consists of sequential sampling of information, with the choice implemented once the decision maker accumulates enough net evidence in favor of one of the options (*Ratcliff et al., 2016*; *Shadlen and Shohamy, 2016*). Furthermore, experimental and theoretical accounts support the idea that such evidence accumulation is a domain-general mechanism underlying judgments about both the objective state of the physical world (perceptual decisions) (*Bogacz et al., 2010*; *Forstmann et al., 2016*; *Gold and Heekeren, 2014*; *Hanks and Summerfield, 2017*; *O'Connell et al., 2018*) and the subjective reward value of different choice options (*Basten et al., 2010*; *Bhatia, 2013*; *Clithero, 2018*; *De Martino et al., 2013*; *Diederich, 2003*; *Fudenberg et al., 2018*; *Gluth et al., 2012*; *Hare et al., 2011*; *Hayden et al., 2011*; *Hunt et al., 2012*; *Hutcherson et al., 2015*; *Krajbich et al., 2015*; *Mormann et al., 2010*; *Philiastides and Ratcliff, 2013*; *Polanía et al.,*

*2014*; *Rodriguez et al., 2014*; *Roe et al., 2001*; *Tajima et al., 2016*; *Trueblood et al., 2014*; *Webb, 2019*; *Woodford, 2014*; *Zhao et al., 2020*).

Visual attention allows us to selectively process the information in our environment, allocating greater computational resources to elements of interest in the visual scene, at the cost of diminishing the processing of unattended components (*Carrasco, 2011*; *Chelazzi et al., 2011*; *Failing and Theeuwes, 2018*; *Itti and Koch, 2001*). Thus, the sequential orienting of attention toward different stimuli is crucial for understanding the whole visual scene and guiding our gaze through it (*Eimer, 2014*). Attention can either be directed overtly (i.e., by eye fixation) or covertly (during constant fixation), but the effects on neural processing and the underlying causal mechanisms are thought to be strongly related (*Moore and Zirnsak, 2017*).

Investigations into the link between overt attention and decision making have shown that, during decision making, subjects shift their gaze between the options until one of them is selected. These findings have led to proposals that overt attention may influence the evidence comparison process that guides choice behavior (*Amasino et al., 2019*; *Ashby et al., 2016*; *Cavanagh et al., 2014*; *Folke et al., 2016*; *Hunt et al., 2018*; *Krajbich et al., 2010*; *Shimojo et al., 2003*; *Towal et al., 2013*; *Glickman et al., 2019*). This proposed mechanism has been formalized by the attentional drift diffusion model (aDDM; *Fisher, 2017*; *Konovalov and Krajbich, 2016*; *Krajbich and Rangel, 2011*; *Krajbich et al., 2010*; *Krajbich, 2019*; *Smith and Krajbich, 2019*; *Tavares et al., 2017*) and is supported by several reports of reliable correlations between gaze patterns and choice (*Kovach et al., 2014*; *Sepulveda et al., 2020*; *Smith and Krajbich, 2018*; *Stewart et al., 2015*; *Vaidya and Fellows, 2015*) as well as experimental manipulations of attention that affect choice (*Armel et al., 2008*; *Colas and Lu, 2017*; *Gwinn et al., 2019*; *Lim et al., 2011*; *Milosavljevic et al., 2012*; *Pärnamets et al., 2015*), but see *Ghaffari and Fiedler, 2018*; *Newell and Le Pelley, 2018*. However, most of the existing studies have manipulated attention by means of salient stimulus changes or direct experimental instruction, which may possibly induce experimenter demand effects on choice.

In the present study, we investigated the possible neural interface between overt attention and value-based choice using non-invasive brain stimulation, which offers a unique opportunity to test the functional contributions of brain areas reported to guide overt attention without altering the experimental setup. We employed transcranial magnetic stimulation (TMS) and targeted the right human frontal eye field (FEF) – a brain region often reported to be activated during the control of both eye movements and selective visual attention (*Corbetta and Shulman, 2002*; *Hung et al., 2011*; *Juan and Muggleton, 2012*; *Marshall et al., 2015*; *Moore and Zirnsak, 2017*; *Schall, 2015*) (Materials and methods). We combined TMS with a value-based choice task, eye tracking, and the aDDM, to investigate how neural excitability modulations in the FEF affect both overt attention and the variables underlying the value-based choice process. Our choice task capitalizes on an important feature of the aDDM, namely that gaze has an amplifying effect on subjective values and so has a stronger effect on decisions between high-value items than low-value items (*Shevlin and Krajbich, 2021*; *Smith and Krajbich, 2019*; *Westbrook et al., 2020*). Thus, to characterize how TMS affects this process with high sensitivity, we employed a task with two overall-value conditions (low-value and high-value). This allowed us to test whether the FEF plays a causal role in the value-comparison process, and whether this effect is indeed stronger for higher-valued items, thereby indicating value amplification rather than just a gaze bias.

## Results

Forty-five subjects participated in the experiment. First, subjects rated 148 different food items on a scale from 0 to 10 based on how much they would like to eat the food, or pressed the space bar for foods they did not want to eat (*Figure 1A*). After the rating task, subjects received inhibitory TMS over the right FEF (n = 23) or control TMS over the vertex (n = 22) (*Figure 1D*). In the second task, immediately after the TMS procedure, subjects made 180 decisions between pairs of positively rated food items (*Figure 1B*). The task had two conditions, differing with respect to overall value (OV = left item value+ right item value). In the high OV condition, subjects had to choose between two very appetitive (highly rated) foods, while decisions in the low OV condition only involved slightly appetitive (low rated) options (*Figure 1C*). This allowed us to test specifically whether any contributions of FEF to value processing during choice were additive (i.e., similar for both OV levels) or multiplicative

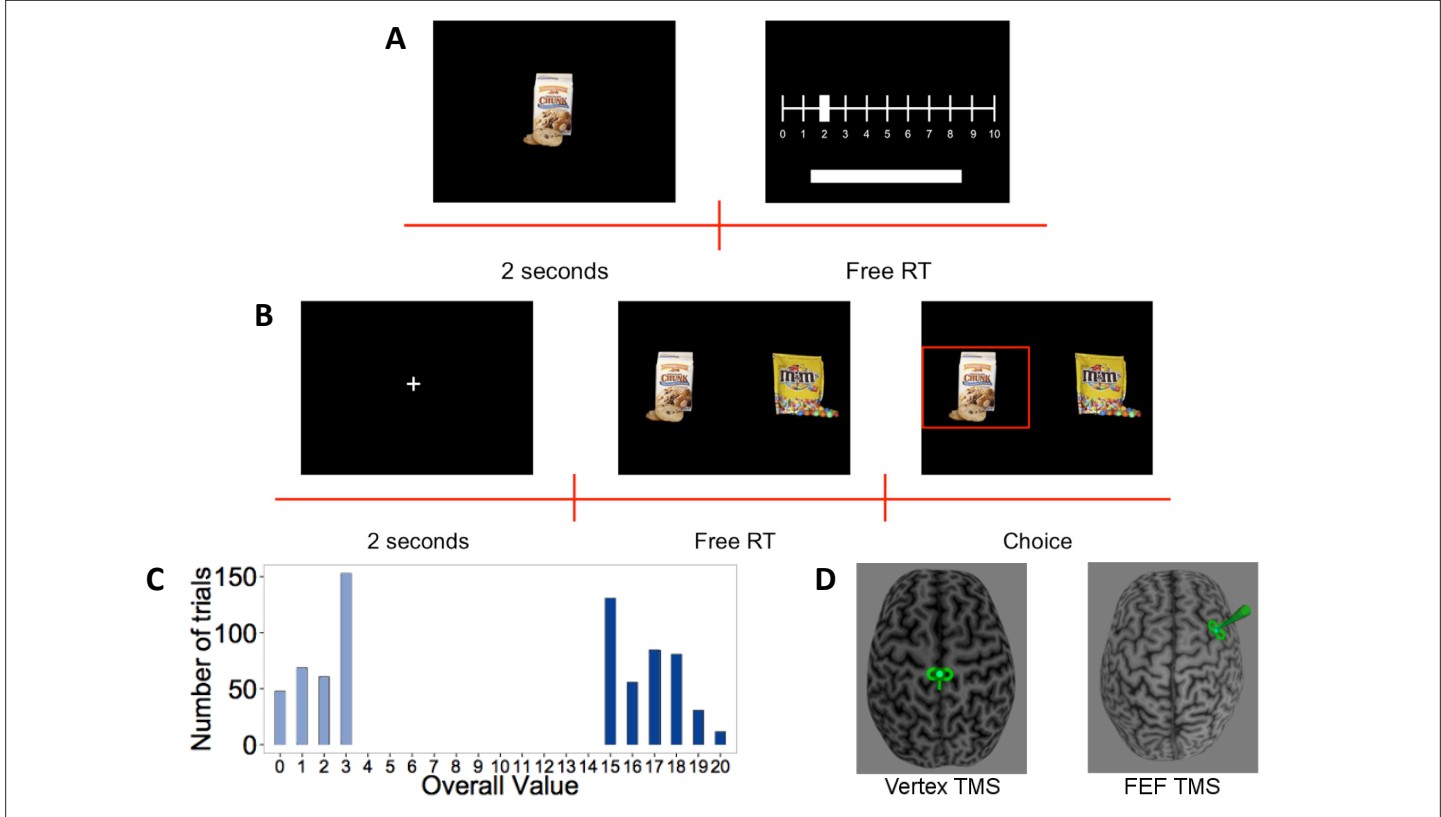

**Figure 1.** Experiment setup. (A) Rating-task timeline: Subjects saw each food item for 2 s and then rated how much they would like to eat it on a scale from 0 to 10, or excluded the item by pressing the space bar (no time limit). (B) Choice-task timeline: Subjects first had to fixate a central cross for 2 s. They then had to choose between the two presented food items using the keyboard. The chosen food was then highlighted for 1 s. (C) Histogram of overall value (OV) in the choice task: Trials were constructed to have either very high or low OV. (D) Stimulation: After the rating task, subjects received continuous theta-burst transcranial magnetic stimulation (TMS) over the vertex (left panel) or right frontal eye field (FEF) (right panel), depicted here schematically by the small green TMS coil symbol over one subject's brain reconstruction.

(i.e., enhanced for high OVs) in nature. During the binary choice task, we recorded subjects' gaze at 250 Hz with an EyeLink-1000 (http://www.sr-research.com/).

## Computational modeling

Our theoretical framework is the aDDM (*Krajbich et al., 2010*, but note that we also demonstrate that our results are robust to the modeling framework). In the standard DDM, decision makers accumulate noisy evidence for the options until the net evidence reaches a predefined boundary. In a value-based DDM, the rate of evidence accumulation ('drift rate') thus reflects the difference in subjective values of the options. The aDDM extends this model by allowing the drift rate to change within a trial, based on which option is fixated. In particular, the model assumes that gaze amplifies the value of the fixated (relative to non-fixated) option, shifting the drift rate toward that choice option and increasing the likelihood that it is chosen. Because of this amplifying (multiplicative) mechanism in the model, gaze has a stronger effect on the drift rate for high-value options, leading to shorter reaction times (RTs) and a larger effect of dwell time on choice probability (*Shevlin and Krajbich, 2021*; *Smith and Krajbich, 2019*; *Westbrook et al., 2020*; *Figure 2*).

Formally, the aDDM captures the evidence accumulation process with a relative decision value (RDV) that evolves stochastically as follows. Let $V_t$ be the value of the RDV at time $t$ while $d$ is a constant that controls the speed of change (in units of $ms^{-1}$), and let $r_{left}$ and $r_{right}$ denote the values of the two options. Let $\theta$ (between 0 and 1) be a weight that discounts the value of the unattended alternative and, therefore, biases the RDV in favor of the attended one. $\xi$ is white Gaussian noise with variance $\sigma^2$, randomly sampled once every millisecond. Then, when a subject fixates on the left option,

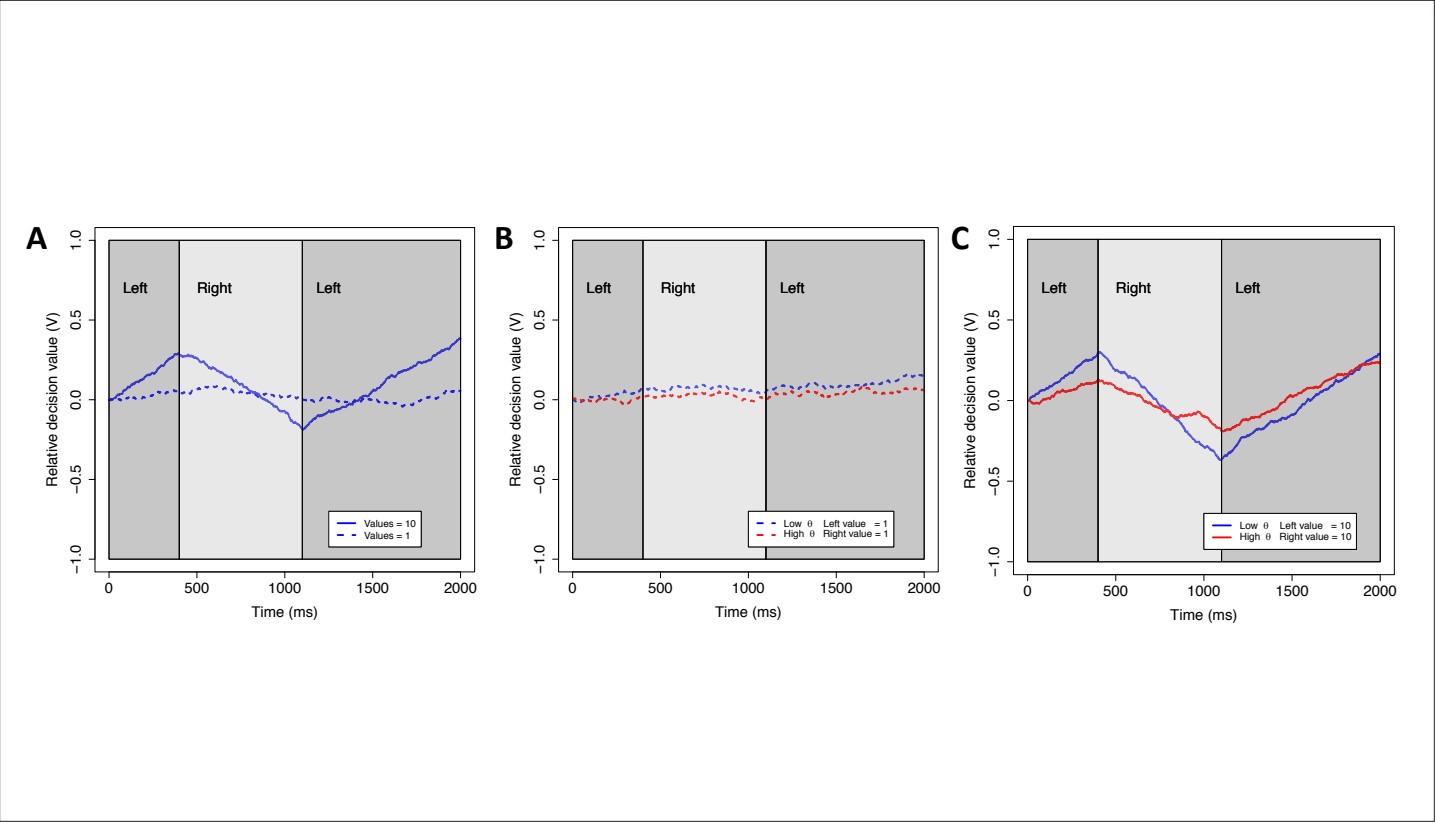

**Figure 2.** Simulations of the attentional drift diffusion model (aDDM). To provide an intuition for why the aDDM makes different predictions for low and high overall value (OV), we simulated the aDDM, once with low values (dashed lines) and once with high values (solid lines). The simulations were run for a subject with a typical gaze discount factor (low $\theta$-value, blue line) ($\theta = 0.3$) and a subject with less of a discount (red line) ($\theta = 0.5$). Dark (light) gray areas indicate periods where the subject is looking at the left (right) item. The relative decision value ($V$) evolves over time with a slope that is biased toward the item that is being fixated. The left item is selected when $V$ reaches an upper boundary and the right item is selected when $V$ reaches a lower boundary. (A) For a given $\theta$, higher OV results in bigger changes in the slope (drift rate) when gaze shifts between left and right. (B) For low OV, shifts in gaze lead to little change in drift rate regardless of $\theta$. (C) For high OV, shifts in gaze lead to large changes in drift rate, particularly for lower $\theta$. Taken together, this means that the behavioral difference between low and high $\theta$ is much more pronounced for high vs. low OV. In particular, larger changes in drift rate lead to faster decisions and a stronger propensity to choose the longest fixated option.

the RDV progresses according to $V_t = V_{t-1} + d\,(r_{left} - \theta\,r_{right}) + \xi$, and when the subject fixates on the right option, the RDV changes according to $V_t = V_{t-1} + d\,(\theta\,r_{left} - r_{right}) + \xi$.

If the RDV reaches the positive boundary the left reward is chosen and if it reaches the negative boundary the right reward is chosen.

The parameter $\theta$ captures the degree to which the value of the fixated option is amplified by attention. A lower value of $\theta$ (closer to 0) indicates a stronger attentional influence, meaning that the decision maker is more likely to choose the longer-attended option and to respond faster (**Figure 2**).

## Does TMS affect the orienting of overt visual attention?

Before turning to the behavioral results, we first investigate whether FEF TMS had any effect on the gaze patterns, that is, the deployment of overt attention. Any such effects would need to be accounted for in our subsequent analyses. Our subjects had a strong tendency to look first at the left food item (69.3%, 95% CI = [62.4%, 76.1%], p = 10$^{-6}$) (**Figure 3**). A logistic regression revealed that this tendency was reduced for high compared to low OV trials (OV: $\beta = -0.194$, CI = [−0.297,−0.091], p = 0.0002) in the vertex group, but that this difference between OV conditions was significantly reduced in the FEF group (FEF × OV: $\beta = 0.222$, CI = [0.058, 0.385], p = 0.008) (see Materials and methods).

A potential explanation for this effect is that for high OV trials, subjects are less likely to look left first because they are more likely to first look at the better item. We tested this idea with another logistic (mixed-effects) regression, looking at the probability of fixating the higher-rated item first.

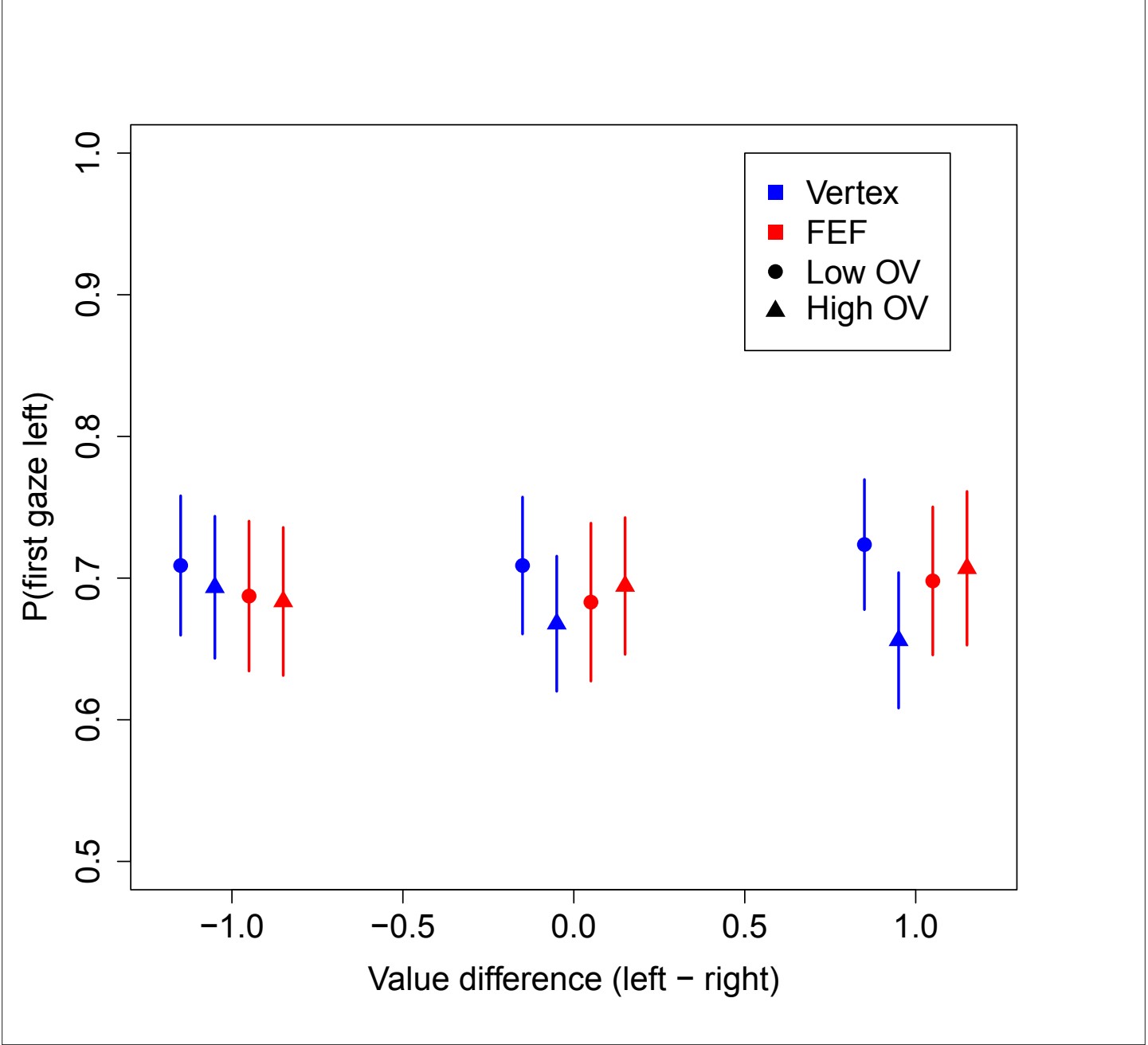

**Figure 3.** Frontal eye field (FEF) effects on gaze patterns. Probability of looking first at the left food item during a trial. Subjects had a strong tendency to look left first. This tendency was slightly reduced in high vs. low overall value (OV) trials for the vertex group but not the FEF group. Bars are s.e.m., and the x-coordinates of the points are jittered around the true values of –1, 0, and 1.

Consistent with prior work (**Krajbich et al., 2010**), a subject's first gaze was no more likely than chance to go to the higher-rated item (50.2%, CI = [49.0%, 51.3%], p = 0.78). Moreover, we found no evidence for any OV or TMS effects. While the probability of fixating the better item was numerically lower for high OV trials in the vertex group (OV: $\beta$ = –0.100, CI = [–0.253, 0.053], p = 0.2) and numerically higher under FEF TMS (FEF × OV: $\beta$ = 0.126, CI = [–0.089, 0.340], p = 0.25), none of these effects reached statistical significance (see Materials and methods).

In sum, the only qualitative effect observed for FEF TMS on first fixation location seems to be that subjects look left first at the same rate in high (69.5%) and low (69.0%) OV trials, compared to vertex TMS for which the rate of looking left first is slightly reduced in high (67.2%) vs. low (71.4%) OV trials.

We account for these effects in our later modeling to ensure that what we observe in the model parameters is not due to these differences in initial gaze allocation.

We also examined whether FEF TMS affected the gaze dwell times. In keeping with prior work (*Krajbich et al., 2010*; *Krajbich et al., 2012*), we separately analyzed first, middle, and last dwells. The basic reasoning is that first dwells tend to be shorter than the rest. The last dwell of the trial is also different from the rest in that it is cut short by the crossing of the decision threshold.

In the regression analyses, we found no hint of any main TMS effects or interaction effects (all p > 0.2) (see Materials and methods). Therefore, we conclude that FEF TMS did not induce any observable changes in dwell times relative to vertex TMS.

## How does TMS affect choices and RTs?

To test the aDDM predictions about how the right FEF should contribute to value-based choices, we compared trials with high or low OV, since the aDDM predicts stronger attentional effects (and therefore TMS modulation) for trials with high OV. Based on our theoretical framework, we tested several specific hypotheses with regard to choices and RTs.

First, the aDDM predicts that subjects should select the longest-fixated alternative, and this effect should be stronger for high-OV trials. If the right FEF play a role in bringing about this multiplicative effect, then the FEF (compared to vertex) group should show a weaker effect of gaze on choice, particularly for the high-OV trials. We tested this hypothesis with a trial-level logistic mixed-effects regression in which the choice of the left item was the dependent variable, as a function of value difference (VD), TMS condition, OV, dwell-time advantage, and the interactions between the last three variables. The results confirmed all the predictions. During low-OV trials, subjects in the vertex group were more inclined to choose the left food item as its dwell-time advantage increased (dwell-time

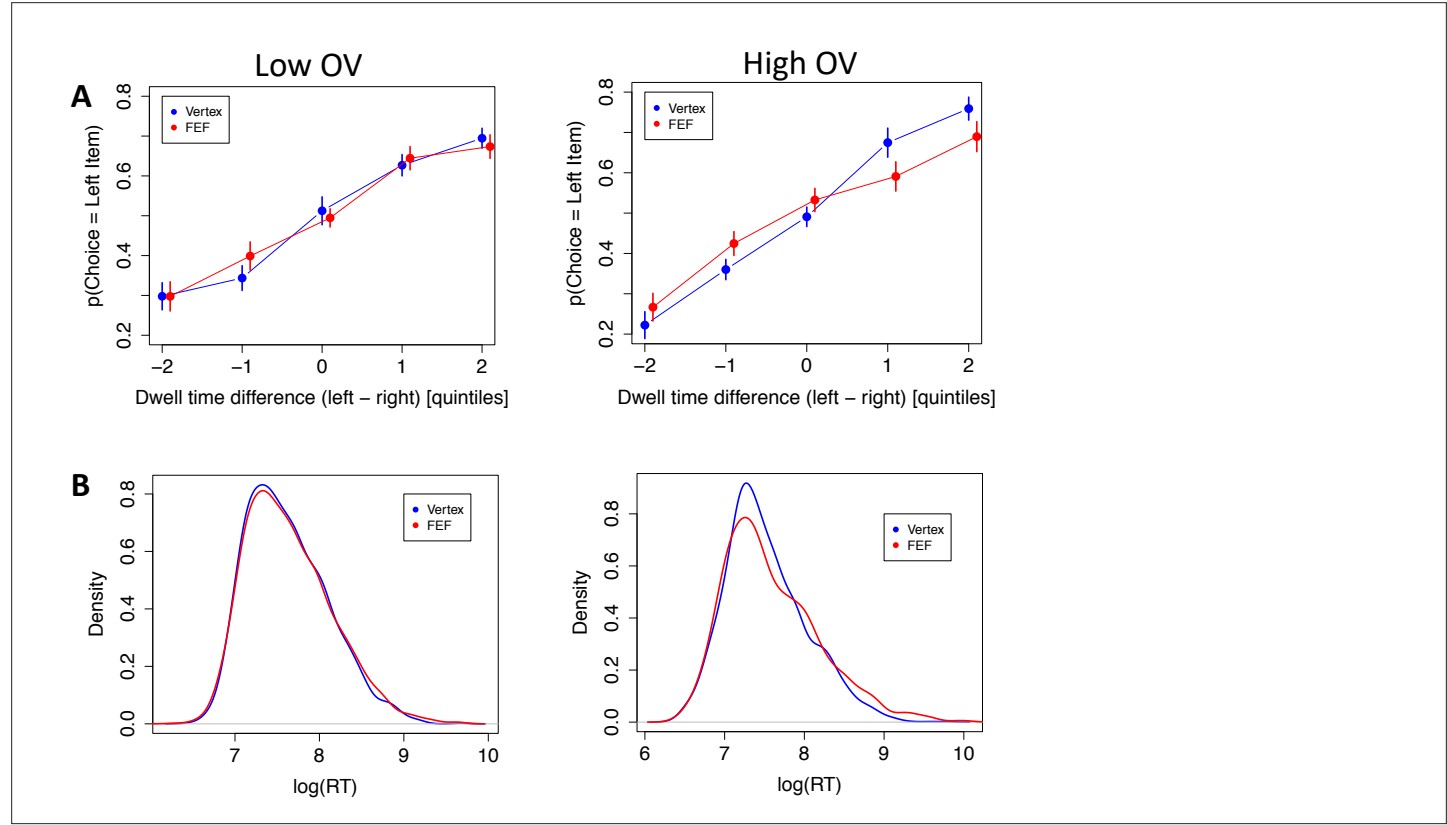

**Figure 4.** Behavioral results. The left panels are for low overall value (OV) trials, the right panels are for high OV trials, and vertex subjects are displayed in blue, frontal eye field (FEF) subjects in red. (A) Choice data: The probability of choosing the left item as a function of the total dwell time difference between the left and right items. Quintiles were determined at the subject level. Quintile 0 represents decisions where both items had similar total dwell times. Negative quintiles indicate more dwell time for the right item and positive quintiles indicate more dwell time for the left item. Bars are s.e.m. (B) Density plots of log reaction time (RT) for low and high OV, respectively.

advantage: $\beta$ = 0.966, CI = [0.477, 1.455], p = 0.0001). This effect increased strongly for high-OV trials, consistent with a multiplicative effect (OV × dwell-time advantage: $\beta$ = 0.622, CI = [0.375, 0.869], p = $10^{-6}$; *Figure 4A*). The FEF group showed no difference with the vertex group for the dwell-time effect in low-OV trials (FEF × dwell-time advantage: $\beta$ = 0.310, CI = [–0.373, 0.993], p = 0.37) but showed a substantial decrease relative to the vertex group in high-OV trials (FEF × OV × dwell-time advantage: $\beta$ = –0.709, CI = [–1.018,–0.401], p = $10^{-5}$), In fact, the difference in dwell-time effects between low and high OV was eliminated in the FEF-TMS subjects, suggestive of a disruption of the multiplicative effect of gaze on value processing of the fixated option (*Figure 4A*, see Materials and methods).

We investigated the robustness of this effect in a few ways. First, we additionally included an interaction between VD, TMS condition, and dwell-time advantage to check whether TMS had an effect on the interaction between VD and gaze. VD showed no significant interaction with dwell-time advantage ($\beta$ = –0.051, CI = [–0.199, 0.096], p = 0.50), with TMS condition ($\beta$ = –0.124, CI = [–0.309, 0.061], p = 0.19), or with their combination ($\beta$ = –0.0001, CI = [–0.188, 0.188], p = 1). Meanwhile, the triple interaction between OV, dwell-time advantage, and TMS condition remained highly significant ($\beta$ = –0.710, CI = [–1.019,–0.401], p = $10^{-5}$). Second, we sought to account for whether subjects exhibited spatial gaze biases, measured by where they tended to look first. We split the subjects into two groups based on whether they more likely to look at the left or right item first. Thirty-five subjects showed a leftward bias, while 10 showed a rightward bias, with a comparable distribution across the two experimental groups (four in the vertex group and six in the FEF group). We ran the mixed-effects logistic regression from the previous paragraph on each group separately. Reassuringly, the key triple interaction between dwell-time advantage, OV, and TMS condition was very similar in magnitude and significant in both groups (leftwards biased subjects: $\beta$ = –0.700, CI = [–1.042,–0.359], p = $10^{-4}$; rightwards biased subjects: $\beta$ = –0.779, CI = [–1.518,–0.040], p = 0.04). Third, we investigated the stability of the effect over the course of the experiment. We again ran the same logistic mixed-effects regression from the previous paragraph, but with an additional interaction between trial number and all the other regressors (and their interactions) except VD. This regression yielded no significant effects involving trial numbers (all p > 0.24), suggesting that TMS effects on behavior did not change systematically across the duration of the experiment.

The aDDM also predicts that subjects should have shorter RTs for high-OV trials compared to low-OV trials. Again, this effect should be reduced for the FEF (relative to vertex) group if FEF TMS reduces the multiplicative value-discounting effect of overt attention on the choice process. We tested this with a trial-level mixed-effects regression in which log(RT) was the dependent variable, as a function of |VD|, TMS condition, OV, and the interaction between the last two variables. We included |VD| as a measure of decision difficulty, as is standard; here, the effect was only marginal (|VD|: $\beta$ = –0.02, CI = [–0.046, 0.007], p = 0.15), presumably because of the very narrow range of |VD| in our task, chosen to enhance sensitivity for our main question of interest. As expected, subjects in the vertex group were faster in high-OV compared to low-OV trials (OV: $\beta$ = –0.1, CI = [–0.149,–0.052], p = 0.0006), and this was not changed by TMS for low-OV trials (TMS: $\beta$ = 0.021, CI = [–0.136, 0.178], p = 0.79). The expected interaction between TMS and high OV was not significant, but pointed in the correct direction and numerically cut the effect of OV on RT by about half (TMS × OV: $\beta$ = 0.06, CI = [–0.019, 0.139], p = 0.15).

For a more sensitive analysis of the RTs, we utilized the Kolmogorov-Smirnov (K-S) method to test for differences between the RT distributions. K-S tests revealed no difference between stimulation groups for low-OV trials ($D$ = 0.022, p = 0.71) but a significant difference for high-OV trials ($D$ = 0.064, p = 0.005) (*Figure 4B*). They also revealed significant differences between high- and low-OV trials in both groups, though the difference was numerically larger for the vertex group ($D$ = 0.097, p = $10^{-8}$) than for the FEF group ($D$ = 0.091, p = $10^{-7}$). Admittedly, however, these tests do not account for repeated measures per subject, so they should be treated with caution.

## Model fitting

According to our hypothesis, inhibitory TMS on the right FEF should decrease the multiplicative gaze discount (i.e., increase $\theta$) on the non-fixated option during the evidence accumulation process. The behavioral analyses reported above are consistent with this hypothesis. Next, we made simultaneous use of choice and RT data, using diffusion modeling, to provide more direct evidence for this

hypothesis, by showing that the estimated $\theta$ parameters were indeed higher for subjects in the FEF-TMS group compared to the vertex-TMS group.

To do so, we used the software package HDDM to fit a hierarchical diffusion model that accounts for the effects of gaze on choice. In addition to the standard threshold separation ($a$), starting-point ($z$), and non-decision time ($T_{er}$) parameters, we also estimated drift rate as a function of the food ratings and dwell-time proportion (see Materials and methods).

Looking first at threshold separation, we found no difference between the groups, either in the group-level parameter distributions ($a_{FEF}$ = 2.67, [2.42, 2.85]; $a_{vertex}$ = 2.63, [2.42, 2.94]; mean difference = 0.042, [–0.337, 0.434], p($a_{vertex}$< $a_{FEF}$) = 0.60) or in the subject-level parameter estimates ($t(40.3)$ = 0.29, p = 0.78). This clearly indicates that there was no change in response caution between the two groups. Looking next at the non-decision-time parameter, we again found no difference between the groups, either in the group-level parameter distributions ($Ter_{FEF}$ = 694 ms, [633, 762]; $Ter_{vertex}$ = 666 ms, [609, 728]; mean difference = 0.029, [–0.073, 0.126], p($Ter_{vertex}$< $Ter_{FEF}$) = 0.74) or in the subject-level parameter estimates ($t(42.8)$ = 0.72, p = 0.47). This indicates that there was also no change in general RT components that are separate from the decision process.

In the model, we also accounted for potential additive effects of gaze on choice (i.e., changes that do not reflect modulation of value evidence but that are constant and independent of item values). We did observe significant additive effects of gaze direction, but importantly, these effects did not differ between the groups, either in the group-level parameter distributions ($\beta_{3FEF}$ = 0.707, [0.416, 1.010]; $\beta_{3vertex}$ = 0.589, [0.337, 0.861]; mean difference = 0.119, [–0.325, 0.579], P($\beta_{3vertex}$< $\beta_{3FEF}$) = 0.73) or in the subject-level parameter estimates ($t(40.1)$ = 0.86, p = 0.40).

Having established that FEF TMS does not affect general, value-independent response processes, we turn to our key test. Comparing the two groups on the $\theta$ estimates, we found that the estimated $\theta$ was lower for the vertex group ($\theta$ = 0.744) than for the FEF group ($\theta$ = 0.902), marginally at the group level (mean difference = 0.160 [–0.141, 0.468], p($\theta_{vertex}$< $\theta_{FEF}$) = 0.89) and significantly at the subject level ($t(42.9)$ = 2.85, p = 0.007). This suggests that the effects on choice behavior induced by TMS on the right FEF were specifically due to the multiplicative aDDM effects, and not due to changes in response caution, non-decision-time, or additive gaze effects.

We also observed some counteracting spatial biases in the modeling results. In the FEF group, we found evidence for a starting-point bias to the left but for a drift-rate bias to the right, relative to the vertex group. The starting point in the FEF group ($z$ = 0.506, [0.493, 0.519]) was greater than in the vertex group ($z$ = 0.491, [0.477, 0.506]) at the group level (mean difference = 0.014 [–0.007, 0.037], p($\theta_{vertex}$< $\theta_{FEF}$) = 0.93) and at the subject level ($t(29.3)$ = 3.64, p = 0.001). However, the drift-rate intercept in the FEF group ($\beta_0$ = –0.018 [–0.056, 0.019]) was lower than in the vertex group ($\beta_0$ = 0.017 [–0.029, 0.061]) at the group level (mean difference = –0.035 [–0.100, 0.034], p($\theta_{vertex}$< $\theta_{FEF}$) = 0.12) and at the subject level ($t(36.3)$ = –3.44, p = 0.001). Please note that because these effects are of opposite direction, they do not have a net biasing effect on behavior.

In sum, the results that we have presented demonstrate that FEF TMS reduces the effect of gaze on choice even when the observed gaze patterns and possible complex spatial biases are accounted for. Thus, FEF TMS has effects on value discounting during choice that are independent of any effects the stimulation may have on gaze patterns or spatial biases themselves.

## Discussion

Based on our computational modeling framework, we developed a paradigm that allowed us to causally manipulate value-based choice by inhibiting the right FEF with TMS. Formal computational modeling using HDDM confirmed that the FEF inhibition reduced gaze effects on choice in high-value trials, that is, it increased the gaze discount factor $\theta$. Importantly, this change was measured after accounting for the limited effects of the FEF stimulation on the gaze patterns themselves, as well as on value-independent decision processes.

Our results provide a neural validation for a central assumption in the aDDM, namely that overt attention amplifies the subjective values of the items, leading to larger effects on choice for higher-valued items (*Smith and Krajbich, 2019*). In designing our experiment, we capitalized on this feature of the model by contrasting low- and high-value trials. We also constrained VD to be as small as possible, because that is when the effects of gaze are most apparent (*Krajbich, 2019*). By showing that FEF stimulation affected choices and RTs in high-value but not low-value trials, we confirmed a

role for right FEF in attention-based value modulation during choice, and further validated the multi-plicative nature of the aDDM (*Westbrook et al., 2020*).

More generally, our results build on a literature documenting the DDM-like neural mechanisms underlying value-based choice. These papers have used EEG (*Polanía et al., 2014*), fMRI (*Basten et al., 2010*; *Gluth et al., 2012*; *Hare et al., 2011*; *Lim et al., 2011*; *Rodriguez et al., 2015*), and their combination (*Pisauro et al., 2017*), to identify neural signatures of evidence accumulation in structures such as the dorsal/posterior medial prefrontal cortex, dorsolateral prefrontal cortex, and intraparietal sulci. Our results suggest that the right FEF may play a critical role in contributing to the resulting integration activity in this set of regions, through implementing a multiplicative value-discounting effect linked to the focus of overt attention.

Our results also suggest some interesting parallels in how overt attention may influence, by means of FEF and its feedback projections, both perceptual and value systems in the human brain. As for perception – where mainly covert rather than overt attention has been studied – previous psychophys-ical studies have shown that, relative to unattended visual stimuli, perceptual sensitivity is enhanced for attended elements of the visual scene (*Barbot et al., 2011*; *Herrmann et al., 2010*; *Montagna et al., 2009*; *Pestilli et al., 2007*; *Pestilli and Carrasco, 2005*). These investigations suggest that attended visual stimuli may have stronger neural representations than unattended ones, and this view has been supported by neurophysiological reports in humans (*Kastner et al., 1998*; *Liu et al., 2005*; *O'Craven et al., 1997*) and non-human primates (*Connor et al., 1997*; *Martínez-Trujillo and Treue, 2002*; *Reynolds et al., 2000*; *Reynolds and Desimone, 2003*). More precisely, these investi-gations have shown that neuronal populations that code for covertly attended locations of the visual scene display enhanced activity, relative to neurons representing unattended locations. Importantly, the FEF is a possible source of these attention-dependent modulations: Investigations combining brain stimulation with neuroimaging techniques have shown that modulations of neuronal activity in the FEF induce top-down modulatory effects on both behavior and neuronal activity in early visual areas that resemble effects of attention (*Moore and Armstrong, 2003*; *Moore and Fallah, 2004*; *Ruff et al., 2006*; *Silvanto et al., 2006*; *Taylor et al., 2007*). Additionally, it has been suggested that attention-dependent behavioral effects are due to a boost in synchronization between FEF and V4 at the gamma-band frequency (*Gregoriou et al., 2009*).

In light of this information, it is interesting that TMS of the FEF also leads to decreasing modulatory effects of attention on the value of non-attended items, even though these effects were observed for overt rather than covert attention as in the perceptual studies. Could these behavioral effects reflect that the FEF may exert similar top-down modulatory effects on value representations? Exten-sive research indicates that such value representations are found in a large network of brain areas including the ventromedial prefrontal cortex, orbitofrontal cortex, ventral striatum, and the posterior parietal cortex (*Bartra et al., 2013*; *Boorman et al., 2009*; *Clithero and Rangel, 2014*; *Cromwell and Schultz, 2003*; *Kahnt et al., 2014*; *Knutson et al., 2001*; *Padoa-Schioppa and Assad, 2006*; *Plassmann et al., 2007*; *Platt and Glimcher, 1999*). Moreover, these value-related BOLD signals are increased when attention is directed to the value (rather than other aspects) of objects (*Grueschow et al., 2015*; *Grabenhorst and Rolls, 2008*), or when participants attend to specific value-relevant aspects of a given stimulus (*Lim et al., 2011*). Interestingly, in another analogy to the perceptual domain, attending to the value of items leads to increases in fronto-posterior synchronization in the gamma frequency (*Polanía et al., 2014*), and decreasing the degree of this coherence by brain stim-ulation leads to inaccurate value-based choices (*Polanía et al., 2015*). None of these studies have explicitly examined to what degree these attentional effects on value-based choices may involve the FEF, but our current results suggest that this should be a fruitful area for future studies. Moreover, given that many of the attentional effects of the FEF in the human brain appear to be lateralized to the right FEF (*Ruff et al., 2009*; *Vernet et al., 2014*), our results also raise the interesting question whether stimulation of the left FEF may have similar or different effects on the value modulations occurring as a consequence of overt attention.

One set of findings that is difficult to fully explain is why the FEF group displayed a relative starting-point bias to the left but a drift-rate bias to the right. The starting point of the DDM is meant to capture pre-decisional information that subjects might incorporate into their choices, while the drift rate is meant to capture information that is gathered while evaluating the options. Both positive starting-point and drift-rate biases increase the probability of choosing left, but they have different effects

on the RT distributions. The two groups did not differ in their propensity to choose the left option, consistent with the starting-point and drift-rate biases pointing in opposite directions. Because these two mechanisms have similar effects on behavior, they can often be negatively correlated when there is no true effect. This is likely what happened here, but these results may warrant further investigation.

Taken together, our findings demonstrate the relevance of the FEF for gaze-dependent modulations of value-based decision processes, and they suggest directions for future investigations on the interaction between visual-attention brain networks and areas coding value signals.

## Materials and methods
### Participants
Forty-five right-handed subjects (20 females, mean age ± SD = 23.14 ± 2.39) without a history of implanted metal objects, seizures, or any other neurological or psychiatric disease participated in the experiment. No power analysis was used but the sample size was based on comparable TMS and eye-tracking studies at the time of data collection. Only subjects who reported not being on a diet were allowed to participate. Subjects were informed about all aspects of the experiment and gave written informed consent. Subjects received monetary compensation for their participation, in addition to receiving – at the end of the experiment – the chosen food item from a randomly selected choice trial. The experiments conformed to the Declaration of Helsinki and the Ethics Committee of the Canton of Zurich approved the experimental protocol.

### Experimental design
In a first task, subjects rated 148 food items (average duration of 10 min and 16 s, SD = 1 min and 5 s). Every food item was presented individually on a computer screen for 2 s, followed by a rating screen (free response time). Subjects were instructed to press the space bar for those food items that they did not like at all, and to rate the remaining items on a scale from 0 to 10 based on how much they would like to eat that food at the end of the experiment. This rating task gave us a measure of the subjective value for each food item and allowed us to exclude disliked items.

After the rating task, subjects received inhibitory TMS (see below) on the right FEF or control stimulation on the vertex. We chose the right FEF because this structure is one part of the well-established 'dorsal attention network' (*Corbetta and Shulman, 2002*) and is known to contain neurons involved in various functions relevant for attention and visual cognition, such as target discrimination, overt attention, saccadic eye movements, visual search, and covert attention toward specific visual field locations (*Moore and Zirnsak, 2017*; *Schall, 2015*). Moreover, FEF neurons have been shown to encode the reward value of objects in the current visual scene (*Ding and Hikosaka, 2006*; *Glaser et al., 2016*; *Roesch and Olson, 2007*; *Serences, 2008*) and are modulated in their attention-guiding function by dopaminergic neuromodulation (*Noudoost and Moore, 2011*). This suggests close functional interactions of the FEF with value-coding and dopaminergic reward systems, making this region an ideal candidate to house brain mechanisms responsible for top-down influences linked to overt attention on value computations during goal-directed choice. Due to the well-established right-hemispheric dominance in humans for the orienting of spatial attention and other visual-cognition functions (*Gitelman et al., 1999*; *Heilman and Van Den Abell, 1980*; *Mesulam, 1981*; *Ruff et al., 2009*; *Vernet et al., 2014*), we chose the right FEF as our target.

Subjects were randomly assigned either to the control stimulation group or to the experimental stimulation group (FEF-TMS) before showing up to the experiment. Subjects were blind to their stimulation site, but the experimenters were not (at any part of the experiment or analysis). In the second task, immediately after the stimulation procedure, participants made 180 decisions between pairs of positively rated food items (average duration of 20 min and 36 s, SD = 5 min and 18 s). The food items we presented were selected such that – for each participant – the difference in ratings between the left and right items (VD = left item value − right item value) was constrained to be –1, 0 or +1. This allowed us to focus on difficult choice problems where overt attention is most likely to affect the choice outcomes and where modulatory effects due to the TMS should thus be most sensitively observed.

Choice trials with no gaze time on any food item were excluded from the analysis (0.008 % of the pooled data from the 45 subjects). The mean (s.e.m.) number of trials dropped per subject was 1.44

± 0.67. Both tasks were programmed in Matlab 2013b (Matworks), using the Psychophysics Toolbox extension (*Brainard, 1997*). We used R for statistical analysis as well as the HDDM analysis package for diffusion modeling (*Wiecki et al., 2013*).

## Transcranial magnetic stimulation

Subjects performed the binary choice task after receiving continuous theta burst TMS on the FEF (experimental group, n = 23) or the vertex (control group, n = 22). In this TMS protocol – known to reduce neural excitability in the targeted area for more than 30 min – 600 magnetic pulses are administered over 40 s in bursts of three pulses at 50 Hz (20 ms) repeated at intervals of 5 Hz (200 ms) (*Huang et al., 2005*). Prior to the experimental tasks, a structural T1-weighted anatomical MRI scan was acquired for every subject and reconstructed in 3D for online neuro-navigation and precise placement of the TMS coil, with the Brainsight system (Rogue Research, Montreal, Canada). TMS pulses were delivered using a biphasic repetitive stimulator (Superapid2, Magstim, Withland, UK) with a 70 mm diameter eight-figure coil, and stimulation intensity was calibrated, for each subject, at 80 % of active motor threshold. Passive motor threshold was defined as the single-pulse intensity required to elicit motor evoked potentials (>200 mV in amplitude) in at least five of ten pulses as indexed by electromyography in the Brainsight system (Rogue Research, Montreal, Canada). Active motor threshold was obtained by having the subject exert constant pressure, at 20 % of maximum force, between the index finger and the thumb during the thresholding procedure described above. To stimulate the right FEF, the center of the coil was located over the right hemisphere, just anterior to the junction between the pre-central sulcus and superior frontal sulcus (MNI coordinates: xyz = 35.6 (±2.5), 3.9 (±2.7), 64 (±2.3)). During the stimulation procedure, the coil was positioned tangential to the targeted site and kept steady with a mechanical holding device (Manfrotto, Cassola, Italy), with its handle oriented ~45° in a rostral-to-caudal and lateral-to-medial orientation (i.e., parallel to the central sulcus). For stimulation on the vertex, the procedure was as above except that the center of the coil was located over the central fissure, at the intersection of the left and right central sulci, with the handle pointing backward.

## Behavioral analysis and model fitting

We first investigated whether TMS had any effect on the gaze patterns in the data. We conducted a logistic regression with clustered standard errors (mixed-effects models would not converge) in which first spatial fixation location (left = 1, right = 0) was the dependent variable, regressed on OV interacted with a dummy variable for FEF-TMS group. We then conducted a mixed-effects logistic regression where first fixation location (coded as higher-rated item = 1, lower-rated item = 0) was regressed on OV interacted with a dummy variable for FEF-TMS group. Finally, we also conducted three regressions with clustered standard errors (mixed-effects models would not converge) to analyze the length of first, middle (all but first and last), and last dwell times. In each regression, log dwell time was regressed on TMS condition, position (left or right), OV, and VD (left – right), all interacted together.

We next tested specific hypotheses about choices and RTs based on our theoretical framework. First, we conducted standard generalized linear model analyses. Specifically, we conducted a trial-level logistic mixed-effects regression in which the chosen item (left = 1, right = 0) was the dependent variable, regressed on VD (left – right), a dummy variable for FEF-TMS group, a dummy variable for the OV condition, dwell-time advantage (total gaze dwell time spent on left – right item), and the interactions between the last three variables. The dwell-time advantage is our variable of interest as it captures the effect of increased attention on choice; VD is included as a control variable to account for the difficulty of the decision. The model with full random effects would not converge, so we report the model with a random intercept and random slopes for VD and dwell-time advantage; we obtain nearly identical results if we instead simply omit the random intercept from the full model. Additionally, we ran the same logistic regression model but with clustered standard errors, which is an alternative way to account for repeated observations within subjects. We additionally ran versions of this same model with interactions between VD and the other variables (except OV), with interactions between trial number and the other variables (except VD), and with two groups of subjects based on whether they were more likely to look left or right first.

To examine differences in RTs, we first computed median RTs in high and low OV trials for each subject and then compared them using paired t-tests. For a more sensitive analysis, we additionally performed non-parametric K-S tests between the pooled RT distributions.

All behavioral analyses were conducted using R and mixed-effects regressions used the R package lme4.

Second, to test for differential effects in how TMS affected attention-based choice mechanisms, we utilized the HDDM package (*Wiecki et al., 2013*). This package uses Bayesian methods to estimate both group-level and subject-level DDM parameters. The package has a very useful regression feature, allowing the user to regress DDM parameters on trial-level features. A recent paper noted that one can use this feature to estimate gaze effects on drift rate (*Cavanagh et al., 2014*). A simple trick allows us to actually recover the gaze-discounting parameter $\theta$ directly using this technique. In particular, we run the following regression:

$$v = \beta_0 + \beta_1 \left(r_{left}g_{left} - r_{right}g_{right}\right) + \beta_2 \left(r_{left}g_{right} - r_{right}g_{left}\right) + \beta_3 \left(g_{left} - g_{right}\right) + \varepsilon$$

where $v$ is the drift rate, $r$ are the values of the options, $g$ is the fraction of the trial spent looking at the option, and the subscripts *left* and *right* indicate the left and right options, respectively. $\beta_1 = \beta_2$ is the special case where gaze has no multiplicative effect on drift rate and $\beta_3 = 0$ is the special case where gaze has no additive effect on drift rate; in that case the model reduces to simply $v = \beta_0 + \beta_1 \left(r_{left} - r_{right}\right) + \varepsilon$, which is the standard DDM. When $\beta_1 > \beta_2$, gaze has an amplifying effect on drift rate, as predicted by the aDDM, and $\beta_2/\beta_1 = \theta$. We include the third term $\beta_3$ to account for any possible additive effects of gaze on choice (*Cavanagh et al., 2014*; *Westbrook et al., 2020*).

For these models, we used the default priors in HDDM. As recommended in the HDDM documentation, we allowed for 5 % of trials to be 'outliers', that is, generated by a non-DDM process. We used three chains of 10,000 samples each, with the first 5000 samples discarded as burn in. To ensure model convergence, Gelman-Rubin r-hat values were inspected and all were less than 1.1.

## Acknowledgements

Christian Ruff and Ernst Fehr jointly supervised this work as shared senior authors. Ernst Fehr acknowledges support from the European Research Council (Advanced Grant 295642-FEP). Christian Ruff acknowledges support from the Swiss National Science Foundation (SNF grants 105314_152891 and 100019L_173248) and the European Research Council (Consolidator Grant COG-2016–725355). Ian Krajbich acknowledges support from the National Science Foundation (NSF Career Award 1554837) and the Cattel Sabbatical Fund. We thank the staff at the SNS lab and the Neuroscience Center Zurich ZNZ for practical support. We also thank Blair Shevlin and Nitisha Desai for assistance with the modeling.

## Additional information

### Funding

| Funder | Grant reference number | Author |
|---|---|---|
| H2020 European Research Council | 295642-FEP | Ernst Fehr |
| Swiss National Science Foundation | 105314_152891 | Christian Ruff |
| H2020 European Research Council | COG-2016-725355 | Christian C Ruff |
| National Science Foundation | 1554837 | Ian Krajbich |
| Cattell Sabbatical Fund | | Ian Krajbich |
| Swiss National Science Foundation | 100019L_173248 | Christian C Ruff |
| H2020 European Research Council | research and innovation program (grant agreement No. 758604) | Rafael Polania |

| Funder | Grant reference number | Author |
|--------|------------------------|--------|
| ETH Zürich | ETH-25 18-2 | Rafael Polania |

The funders had no role in study design, data collection and interpretation, or the decision to submit the work for publication.

### Author contributions

Ian Krajbich, Conceptualization, Formal analysis, Investigation, Methodology, Supervision, Visualization, Writing – original draft, Writing – review and editing; Andres Mitsumasu, Conceptualization, Data curation, Formal analysis, Investigation, Methodology, Visualization; Rafael Polania, Data curation, Writing – review and editing; Christian C Ruff, Conceptualization, Funding acquisition, Project administration, Supervision, Writing – original draft, Writing – review and editing; Ernst Fehr, Conceptualization, Funding acquisition, Project administration, Supervision, Writing – review and editing

### Author ORCIDs

Ian Krajbich ⬤ http://orcid.org/0000-0001-6618-5675
Rafael Polania ⬤ http://orcid.org/0000-0002-6176-6806
Christian C Ruff ⬤ http://orcid.org/0000-0002-3964-2364

### Ethics

Human subjects: The experiments conformed to the Declaration of Helsinki and the Ethics Committee of the Canton of Zurich approved the experimental protocol.

### Decision letter and Author response

Decision letter https://doi.org/10.7554/eLife.67477.sa1
Author response https://doi.org/10.7554/eLife.67477.sa2

---

## Additional files

### Supplementary files

• Transparent reporting form

### Data availability

All data generated or analysed during this study are included in the manuscript and supporting files. Source data files have been provided for Figures 3 and 4.

The following dataset was generated:

| Author(s) | Year | Dataset title | Dataset URL | Database and Identifier |
|-----------|------|---------------|-------------|-------------------------|
| Mitsumasu A, Krajbich I, Polania R, Ruff C, Fehr E | 2021 | A causal role for the right frontal eye fields in value comparison | https://osf.io/xmkzs/ | Open Science Framework, osf.io/xmkzs/ |

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
