## [Editor Report]

Previous studies have suggested that the fixation of an object enhances the gain of its value signals that are temporally integrated during deliberation. The authors of the present study demonstrated that inhibition of the right frontal eye field (FEF) with transcranial magnetic stimulation reduces this multiplicative effect of fixation, suggesting that the FEF might be involved in the gaze-dependent modulation of value signals during decision making.

---

## [Decision Letter]

**Decision letter after peer review:**

Thank you for submitting your article "A causal role for the right frontal eye fields in value comparison" for consideration by *eLife*. Your article has been reviewed by 3 peer reviewers, including Daeyeol Lee as Reviewing Editor and Reviewer #1, and the evaluation has been overseen by Tirin Moore as the Senior Editor. The following individuals involved in review of your submission have agreed to reveal their identity: Laurence Tudor Hunt (Reviewer #2); Woo-Young Ahn (Reviewer #3).

Essential revisions:

Mitsumasu et al., tested the predictions from the attentional drift-diffusion model (aDDF) in humans by applying "inhibitory" transcranial magnetic stimulation over the right frontal eye field during a value-based binary choice task. The main prediction tested in this study is that the multiplicative effect of gaze on fixated item (quantified by the parameter theta that discounts the effect of value of non-fixated item) would be reduced especially for the high-value items by the inhibition of the FEF via its role on attention. The authors have controlled for any potential direct effect of FEF modulation on eye movements, and still demonstrated the predicted effect of TMS on theta. The experiments and analyses are also carefully conducted.

1) The use of the term attention in this paper (as well as in the previous work) causes some confusion because contrary to the vast behavioral and neurophysiological literature on attention, attention in this study refers to the fixation (hence referred to as "overt" attention by the authors). It might be helpful to clarify this in the introduction and abstract by explicitly stating that the authors are only referring to 'overt attention' (i.e. fixation) thoughout the paper, and are not considering the role of covert attention.

2) The main finding in this study is that the inhibitory effect of TMS reduces the discounting of the non-fixated item. Although this is consistent with the predictions of the aDDM model, it is not obvious how this can be reconciled with the type of attentional modulation observed physiologically, because most FEF neurons are lateralized and there is evidence for TMS to FEF producing lateralised effects in spatial orienting (e.g. https://academic.oup.com/cercor/article/17/2/391/317872). How do the authors account for the fact that the effect of TMS observed in this study in relation to the attentional modulation is not lateralized? It is unfortunate that the study does not include the condition in which the left FEF was stimulated, because this would have served as a very useful control condition.

3) As can also be seen from main figure 4a, the effect size (i.e. the actual degree to which FEF TMS reduces the effect of dwell time difference on choice) is comparatively small. The effect on reaction times is similarly a modest one. This is certainly not unusual in TMS studies, but it does affect the degree to which one can draw strong conclusions about the role of FEF in affecting the multiplicative term of the aDDM (as opposed to, say, a more general role in attention). The authors attempt to rule out other non-specific effects on attention, but this is done slightly informally (as opposed to via model comparison with a competing alternative model to the aDDM) and its limitation should be discussed/acknowledged explicitly.

4) The authors did not study the interaction between value difference and attention, and the impact of FEF TMS on this. This might be another major area where predictions of the aDDM could be tested, but the authors instead focussed on a limited range of value difference trials (those close in value, where gaze time effects might be strongest). It would be good, at least, if the authors could unpick this further.

*Reviewer #1 (Recommendations for the authors):*

Mitsumasu et al., tested the predictions from the attentional drift-diffusion model (aDDF) in humans by applying "inhibitory" transcranial magnetic stimulation over the right frontal eye field during a value-based binary choice task. The main prediction tested in this study is that the multiplicative effect of gaze on fixated item (quantified by the parameter theta that discounts the effect of value of non-fixated item) would be reduced especially for the high-value items by the inhibition of the FEF via its role on attention. The authors have controlled for any potential direct effect of FEF modulation on eye movements, and still demonstrated the predicted effect of TMS on theta. The experiments and analyses are also carefully conducted.

The use of the term attention in this paper (as well as in the previous work) causes some confusion because contrary to the vast behavioral and neurophysiological literature on attention, attention in this study refers to the fixation (hence referred to as "overt" attention by the authors). Confusion caused by this terminology becomes a bit more serious in this study, as the authors begin to study the neuronal mechanisms of the attentional modulation in the FEF. For example, the main finding in this study is that the inhibitory effect of TMS reduces the discounting of the non-fixated item. Although this is consistent with the predictions of the aDDM model, it is not obvious how this can be reconciled with the type of attentional modulation observed physiologically, because most FEF neurons are lateralized. How do the authors account for the fact that the effect of TMS observed in this study in relation to the attentional modulation is not lateralized? Similarly, it is unfortunate that the study does not include the condition in which the left FEF was stimulated, because this would have served as a very useful control condition.

1. Was the left-ward bias with the first gaze consistent across all subjects? If not, did the results from the subjects with the right-ward bias show anything different?

2. Given that the duration of experiment is close to the expected duration of the TMS effect, does the attentional effect of TMS also diminish during the experiment?

*Reviewer #2 (Recommendations for the authors):*

This work aims to test the role of the frontal eye fields in the contribution of attention to value-based decision making. The authors apply offline transcranial magnetic stimulation (TMS) to the frontal eye fields (compared to a control condition where the vertex is stimulated), and perform a classic experiment in which subjects choose between different snack foods while freely gazing between them. The authors find that there is a (previously documented) effect of the overall value of the effects of attention on choice; when the overall value is higher, then dwell time has a greater influence on which item will be chosen. When the FEF is stimulated with TMS beforehand, however, this effect of dwell time on the chosen item is reduced, and there is also an effect on reaction time distributions for this condition. This suggests that FEF contributes to the influence of attention on value-based choice, and can be described by a reduction in the 'discounting' of the attended item in an attentional drift diffusion model.

Strengths:

This work benefits from having clearly described hypotheses and a rich body of related work from which it can directly make clear predictions of how TMS delivered to FEF may affect the interaction of attention and choice. The main effect of interest (main figure 4a) is clearly statistically significant, and is certainly of interest to the field. It is well described by the aDDM model, and the findings are clearly explained and elegantly fit with the central predictions of the experiment.

Weaknesses:

On the other hand, it can also be seen from main figure 4a that the effect size (i.e. the actual degree to which FEF TMS reduces the effect of dwell time difference on choice) is comparatively small. The effect on reaction times is similarly a modest one. This is certainly not unusual in TMS studies, but it does affect the degree to which one can draw strong conclusions about the role of FEF in affecting the multiplicative term of the aDDM (as opposed to, say, a more general role in attention). The authors attempt to rule out other non-specific effects on attention, but this is done slightly informally (as opposed to via model comparison with a competing alternative model to the aDDM).

I was also surprised that the authors did not study the interaction between value difference and attention, and the impact of FEF TMS on this. It seemed like this might be another major area where predictions of the aDDM could be tested, but the authors instead focussed on a limited range of value difference trials (those close in value, where gaze time effects might be strongest). It would be good, at least, if the authors could unpick this further.

It is somewhat a matter of taste, but I think figure 2 would be clearer if the panels held theta constant and varied overall value (that is, the two red lines from a and b were plotted on the same plot, and the two blue lines plotted on another plot). This is because the main point made in the text is that gaze has a stronger effect on the drift-rate for high value options – and so it would help if figure 2 made this point as clearly as possible. However, I can also see why the authors plotted this figure the way they did, so I leave it up to them to decide whether this plot should be rearranged.

I also wondered if it might help to directly illustrate this figure with theta of 0.776 (vertex theta) and 0.937 (FEF theta) rather than just an arbitrary low or high theta. The authors could then refer back to this figure when they fit the model to the data, explaining that these were the values that they used to generate this figure.

I also couldn't quite understand why the authors were only predicting an effect of TMS on the interaction between overall value and dwell-time, but not an effect of TMS on the interaction between value difference and dwell-time. The authors deliberately choose items that have a small value difference precisely because this is where dwell-time effects are strongest, and yet there is still a massive effect of value difference on choice behaviour (Z-statistic of 13.12, from table S1). So shouldn't the authors test the VD*dwelltime*TMS interaction also here?

p.13 "If anything, the probability of fixating the better item was numerically lower for high OV trials in the Vertex group…" – this sentence should be reworded to clarify that neither of the effects described are statistically meaningful.

p.16 I am unclear on how/why the t-tests in the Results section appear to have non-integer degrees of freedom? (e.g. "(t(40.2) = 0.30, p = 0.77)")

p.16 Vertex group are missing confidence intervals ((a = 2.63, CI = []))

*Reviewer #3 (Recommendations for the authors):*

Mitsumasu et al. investigated the role of frontal eye field (FEF) by using TMS, a value-based choice task, eye-tracking, and computational modeling. Authors recruited 45 participants (23 in the TMS experimental group, 22 in the Vertex control group) and have them complete a food choice task where they had to choose between the two presented food items. Authors used logistic regression for behavioral analysis and attentional drift diffusion modeling (aDDM) to estimate the attentional effect (theta parameter). With logistic regression analysis, authors showed that the probability of choosing an item is affected by overt attention (dwell time) in the high overall value (OV) condition as expected (Figure 4) and the theta parameter was significantly higher for the TMS group, which suggests that inhibitory TMS on the right FEF affects the comparison process.

1. Pdf page 11 and 12 – I'm not sure how the equations in page 11 (RDV) and page 12 (drift rate) are combined together for model fitting. Consider sharing the HDDM code as well for replication.

2. pdf page 11 – please indicate which package authors used for behavioral analysis. Also, the models authors used seem confusing and it will be great if exact commands or codes will be provided for replication.

3. Pdf page 12 – Please provide details on hierarchical Bayesian modeling including priors, the number of chains, MCMC samples, convergence check, etc.

4. Authors should not use a t-test and p-values on individual model parameters (I assume posterior means?) for comparing conditions or groups because they are estimated with hierarchical Bayesian modeling. Just use a Bayesian way and compare group parameters (e.g., Kruschke (2014)).

---

## [Author Response]

1) The use of the term attention in this paper (as well as in the previous work) causes some confusion because contrary to the vast behavioral and neurophysiological literature on attention, attention in this study refers to the fixation (hence referred to as "overt" attention by the authors). It might be helpful to clarify this in the introduction and abstract by explicitly stating that the authors are only referring to 'overt attention' (i.e. fixation) thoughout the paper, and are not considering the role of covert attention.

We thank the reviewers for raising this important point. We fully agree that the concept of “attention” is multi-faceted and that we should narrow down the focus of our paper to “overt attention”. We now emphasize in the abstract, introduction, and discussion that we only investigate overt attention (i.e., gaze) and do not consider the role of covert attention.

2) The main finding in this study is that the inhibitory effect of TMS reduces the discounting of the non-fixated item. Although this is consistent with the predictions of the aDDM model, it is not obvious how this can be reconciled with the type of attentional modulation observed physiologically, because most FEF neurons are lateralized and there is evidence for TMS to FEF producing lateralised effects in spatial orienting (e.g. https://academic.oup.com/cercor/article/17/2/391/317872). How do the authors account for the fact that the effect of TMS observed in this study in relation to the attentional modulation is not lateralized? It is unfortunate that the study does not include the condition in which the left FEF was stimulated, because this would have served as a very useful control condition.

We also thank the reviewers for this point. We agree that in the monkey brain, FEF neurons are clearly lateralized, so that one could expect symmetrical effects on performance in the TMS-contralateral hemifield. However, we would like to note that in the human brain, this lateralization is much less clear-cut, probably not symmetric, and may depend strongly on the specific cognitive context (Ruff et al. 2009; Gitelman et al. 1999; Vernet et al. 2014). Even in the study that the reviewers helpfully mention, the lateralization is not obvious, since that study only examined neural activity and performance for TMS-contralateral visual stimulation; it is thus unclear how TMS may affect processing of ipsilateral stimuli. Most crucially, however, it should be noted that FEF neurons have multi-faceted functions that are likely to depend strongly on cognitive context (Vernet et al. 2014). This makes it unlikely that any effects observed for covert attention in the peripheral hemifield would equally apply to a setting requiring overt attention as studied here (see also the previous point raised by the reviewers). We now explicitly discuss this issue in the revised manuscript (p. 5)

The reviewer mentions that left-FEF TMS may have served as a good control condition. While we agree that running such an experiment may be interesting in the future, it is unclear what exactly one may be able to predict for its outcome. On the one hand, several studies have suggested that in the human brain, the left FEF does not have similar visuo-spatial functions as the right FEF and that corresponding stimulation may have weaker effects on performance in visual (but not motor) tasks (Ruff et al. 2009; Vernet et al. 2014). However, since it is well possible that the function we report in our manuscript is distinct from those investigated in the previous FEF TMS studies, it is in fact an open question whether left FEF TMS would have been a good control condition or would reveal similar effects as those reported in the manuscript. We thus highlight this interesting question for future studies in the revised manuscript (see p. 18).

References

Ruff, C. C., Blankenburg, F., Bjoertomt, O., Bestmann, S., Weiskopf, N., and Driver, J. (2009). *Journal of Cognitive Neuroscience*, *21*(6), 1146.

Gitelman, D. R., Nobre, A. C., Parrish, T. B., LaBar, K. S., Kim, Y. H., Meyer, J. R., et al. (1999). *Brain* 122, 1093.

Vernet M, Quentin R, Chanes L, Mitsumasu A, Valero-Cabré A. (2014). *Frontiers in Integrative Neuroscience*. 8, 66.

3) As can also be seen from main figure 4a, the effect size (i.e. the actual degree to which FEF TMS reduces the effect of dwell time difference on choice) is comparatively small. The effect on reaction times is similarly a modest one. This is certainly not unusual in TMS studies, but it does affect the degree to which one can draw strong conclusions about the role of FEF in affecting the multiplicative term of the aDDM (as opposed to, say, a more general role in attention). The authors attempt to rule out other non-specific effects on attention, but this is done slightly informally (as opposed to via model comparison with a competing alternative model to the aDDM) and its limitation should be discussed/acknowledged explicitly.

We respectfully disagree with the reviewers’ estimation that the difference in the effect of dwell-time on choice between conditions is “small” compared to other effects reported for comparable tasks. The differences shown in Figure 4a (High OV) are on the order of 5-10%, which is substantial for a binary-choice study. Moreover, the low p-value of the effect (p=0.002 with clustered errors, p=10^-5^ with mixed effects) also suggests that the effect is not small relative to the noise in the data.

With regards to the specificity of the mechanism, we would like to emphasize that our experiment was explicitly designed to distinguish between the multiplicative aDDM effects and other more general (“attentional”) biases induced by gaze. The finding that TMS conditions differ for High but not Low OV provides clear evidence for the multiplicative mechanism, and this finding is established with a formal test based on the interaction between the TMS condition, OV, and dwell time advantage. That is, the difference in the effect of dwell time on choice between High and Low OV is significantly different between the two TMS conditions (p=10^-5^).

To make this specificity even clearer, we have performed another formal modeling exercise using the DDM and found that the multiplicative (and not additive) mechanism is what differs most between the TMS conditions. It is important to note that the purely additive model is nested within this more general model that we have estimated, so we have in fact provided a formal model comparison. However, with the benefit of hindsight, we appreciate that this may not have been obvious from the way we presented the modelling procedures and the results. To address this point (as well as some of the reviewers’ specific comments) we have now gone into more detail on the HDDM fitting and the resulting distributions of the parameters, the quality of the fits, etc. We hope this now clarifies that FEF-TMS primarily affects the multiplicative term.

4) The authors did not study the interaction between value difference and attention, and the impact of FEF TMS on this. This might be another major area where predictions of the aDDM could be tested, but the authors instead focussed on a limited range of value difference trials (those close in value, where gaze time effects might be strongest). It would be good, at least, if the authors could unpick this further.

As correctly noted by the reviewer, we indeed focused our design on those close-value trials that allowed us to test our hypotheses about TMS effects with the greatest sensitivity. Nevertheless, we agree that it may be interesting to consider possible interactions of value difference and gaze. To address this point within our design, we have now added VD interactions as a regressor to our critical choice regression analysis (with the additional regressors dwell-time advantage, OV, and TMS condition). We find no significant interactions between VD and either of these variables (or their combination), and we now report this in the manuscript on p.13. Second, we have now included a note in the methods and the discussion explaining why we designed our experiment to only include the smallest value differences (to maximize sensitivity for our main effect of interest).

Reviewer #1 (Recommendations for the authors):1. Was the left-ward bias with the first gaze consistent across all subjects? If not, did the results from the subjects with the right-ward bias show anything different?

Overall the probability of fixating the left item first was 69.3%. 10 out of 45 subjects did not show this leftward bias, with a comparable distribution across the two experimental groups (4 in the Vertex group and 6 in the FEF group).

Ten subjects is too small a group to implement formal statistical comparisons, but we nevertheless ran our critical logistic regression of choice on value difference and the interaction between TMS condition, dwell time difference, and OV, separately for this group and the remaining 35 subjects. Reassuringly, the results of the two regressions were very similar. In the left-bias subjects, the dwell-time coefficient was β = 1.091, p = 0.0005, the dwell-time x OV coefficient was β = 0.538, p = 0.0001, and the dwell-time x OV x TMS coefficient was β = –0.700, p = 10^-4^. In the right-bias subjects, the dwell-time coefficient was β = 0.533, p = 0.07, the dwell-time x OV coefficient was β = 1.00, p = 0.001, and the dwell-time x OV x TMS coefficient was β = –0.779, p = 0.04.

We now describe these results in the main text on p. 13.

2. Given that the duration of experiment is close to the expected duration of the TMS effect, does the attentional effect of TMS also diminish during the experiment?

We have checked this in two ways using our critical logistic regression of choice on value difference and the interaction between TMS condition, dwell time difference, and OV. First, we added trial as an additional interaction factor. This regression yielded no significant effects for the main effect and interaction terms of trial number (all p>0.24). Second, we split the data in half (first 90 trials vs. last 90 trials) and ran the original regression (without trial number included) in each of the two halves of the data. The results of the two regressions were very similar. In the first half, the dwell-time coefficient was β = 0.965, p = 0.0002, the dwell-time x OV coefficient was β = 0.735, p = 10^-5^, and the dwell-time x OV x TMS coefficient was β = –0.708, p = 0.001. In the second half, the dwell-time coefficient was β = 1.14, p = 10^-5^, the dwell-time x OV coefficient was β = 0.442, p = 0.02, and the dwell-time x OV x TMS coefficient was β = –0.656, p = 0.006.

We now briefly mention these results in the paper (p. 13). For the sake of conciseness, in the revision we only briefly report and discuss the lack of time effects suggested by the trial-number regression, without reporting the separate split-half analyses. However, we would be happy to include everything if the review team deems this necessary.

Reviewer #2 (Recommendations for the authors):It is somewhat a matter of taste, but I think figure 2 would be clearer if the panels held theta constant and varied overall value (that is, the two red lines from a and b were plotted on the same plot, and the two blue lines plotted on another plot). This is because the main point made in the text is that gaze has a stronger effect on the drift-rate for high value options – and so it would help if figure 2 made this point as clearly as possible. However, I can also see why the authors plotted this figure the way they did, so I leave it up to them to decide whether this plot should be rearranged.

We thank the reviewer for this excellent suggestion. Because the key point of the paper is that Vertex and FEF subjects have different theta values, we do feel it is important to contrast different theta values within the figure. However, we do agree that it is also useful to contrast the predictions with different OV levels. Thus, we have now included an additional panel to display that comparison.

I also wondered if it might help to directly illustrate this figure with theta of 0.776 (vertex theta) and 0.937 (FEF theta) rather than just an arbitrary low or high theta. The authors could then refer back to this figure when they fit the model to the data, explaining that these were the values that they used to generate this figure.

We thank the reviewer for this suggestion but we would prefer to stick with the existing simulation parameters, for a few reasons:

(1) This figure is meant to illustrate the intuition behind the predictions, not the actually observed data. The simulations use a much smaller noise term compared to the best-fitting model, in order to clearly visualize the change in slopes that correspond to shifts in gaze. So, in any case, the plots do not reflect the “true” parameters but are meant to help the reader understand the experimental logic.

(2) The figure was created based on our expectations going into the study. In past research using similar food-choice paradigms, the best estimate of baseline behavior has been θ = 0.3 (Krajbich et al. 2010; Krajbich and Rangel 2011). From that baseline we estimated a modest increase in θ to 0.5. Therefore, this figure accurately represents our prior hypotheses going into the study, which we should not change upon observing the results of our study.

(3) Simulations using the model parameters from the best-fitting model do not appreciably differ from those already in the figure. Author response image 1 is an example of a simulation with θ = 0.78 and 0.94, an additive effect of 0.62 (the average pooled effect), and otherwise the same parameters as the aDDM simulations in Figure 2. We thus do not think the manuscript would gain much from changes to this figure.

**Author response image 1. sa2fig1:** 

I also couldn't quite understand why the authors were only predicting an effect of TMS on the interaction between overall value and dwell-time, but not an effect of TMS on the interaction between value difference and dwell-time. The authors deliberately choose items that have a small value difference precisely because this is where dwell-time effects are strongest, and yet there is still a massive effect of value difference on choice behaviour (Z-statistic of 13.12, from table S1). So shouldn't the authors test the VD*dwelltime*TMS interaction also here?

This is an interesting point. However, since we deliberately limited our design to the smallest value differences (0 or 1) where we expected to see the largest gaze effects, we are not well positioned to address this question. Nevertheless, we ran an expanded version of the logistic regression of left-choice on VD, TMS condition, OV, dwell-time advantage. This time we also included an interaction between VD, TMS condition and dwell-time advantage. While VD continued to be a very significant predictor of choice (p<10^-16^), it showed no significant interaction with dwell-time advantage (p = 0.50), with TMS condition (p = 0.19), or with their combination (p = 1). Meanwhile, the triple interaction between OV, dwell-time advantage, and TMS condition remained highly significant (p = 10^-5^). The results of this analysis are now included in the text (p. 13) and we also discuss more explicitly why we designed our study to focus on trials with small VDs (to increase our sensitivity for the main effects of interest, p. 5).

p.13 "If anything, the probability of fixating the better item was numerically lower for high OV trials in the Vertex group…" – this sentence should be reworded to clarify that neither of the effects described are statistically meaningful.

We have clarified that neither of these effects was significant.

p.16 I am unclear on how/why the t-tests in the Results section appear to have non-integer degrees of freedom? (e.g. "(t(40.2) = 0.30, p = 0.77)")

This is due to Welch’s correction factor for two-sample t-tests with unequal sample sizes and variances.

p.16 Vertex group are missing confidence intervals ((a = 2.63, CI = []))

Thanks. We have added in the missing confidence interval.

Reviewer #3 (Recommendations for the authors):1. Pdf page 11 and 12 – I'm not sure how the equations in page 11 (RDV) and page 12 (drift rate) are combined together for model fitting. Consider sharing the HDDM code as well for replication.

Thanks for pointing out this potential source of confusion. The first set of equations describe the data-generating process, while the second equation is what we actually used to estimate the model. As described immediately above that equation, HDDM has a feature where you can specify the drift rate as a function of trial-level parameters. We input trial-level values and gaze proportions and HDDM estimates β1 and β2, which we then use to calculate θ.

As suggested, we have now included our HDDM code with the other data and code, with our revision, to help clarify these analyses and to allow others to easily replicate them.

2. pdf page 11 – please indicate which package authors used for behavioral analysis. Also, the models authors used seem confusing and it will be great if exact commands or codes will be provided for replication.

We have now clarified that we used R for all behavioral analyses and the package lme4 for mixed-effects models. We have included the R code for all figures and regressions along with the other data and code, with our revision.

3. Pdf page 12 – Please provide details on hierarchical Bayesian modeling including priors, the number of chains, MCMC samples, convergence check, etc.

We have now provided information on the priors, number of chains, and convergence checks in the Methods (p. 10) and in the HDDM notebook.

4. Authors should not use a t-test and p-values on individual model parameters (I assume posterior means?) for comparing conditions or groups because they are estimated with hierarchical Bayesian modeling. Just use a Bayesian way and compare group parameters (e.g., Kruschke (2014))

We thank the reviewer for this comment. We used tests of individual parameters to align with the non-Bayesian behavioral analyses presented earlier in the text. However, we agree that it is also informative to use the Bayesian approach and compare the posterior distributions of the group parameters. Those results are described below and are also now included in the Results (pp. 15-16). In the process, we also improved the model so that responses were coded as left vs. right rather than correct vs. error. This allows us to capture any spatial biases towards the left or right items. The results are qualitatively the same as before, though as seen below, there is a credible difference in starting-point bias and drift-rate intercept between the two groups, where FEF subjects were more likely to be biased towards the left items in starting point but towards the right items in drift rate. Given that these effects point in opposite directions, and that there are thus no overall differences in choice biases towards the left or right items, we don’t dwell on these results in the paper.

In terms of the difference in group θ parameters, the mean difference is 0.160 with a 95% HDI of [–0.141, 0.468] and P(θ_Vertex_< θ_FEF_) = 0.89.

In terms of the additive gaze effect, the mean difference is 0.119 with a 95% HDI of [–0.325, 0.579] and P(β3_Vertex_< β3_FEF_) = 0.73.

In terms of the boundary separation, the mean difference is 0.042 with a 95% HDI of [–0.337, 0.434] and P(a_Vertex_< a_FEF_) = 0.60.

In terms of the non-decision time, the mean difference is 0.029 with a 95% HDI of [–0.073, 0.126] and P(Ter_Vertex_< Ter_FEF_) = 0.74.

In terms of the starting-point bias, the mean difference is 0.014 with a 95% HDI of [–0.007, 0.037] and P(z_Vertex_< z_FEF_) = 0.93.

In terms of the drift-rate intercept, the mean difference is –0.035 with a 95% HDI of [–0.100, 0.034] and P(β0_Vertex_< β0_FEF_) = 0.12.